# *CASP1* Gene Polymorphisms and *BAT1-NFKBIL-LTA-CASP1* Gene–Gene Interactions Are Associated with Restenosis after Coronary Stenting

**DOI:** 10.3390/biom12060765

**Published:** 2022-05-31

**Authors:** Gilberto Vargas-Alarcón, Julian Ramírez-Bello, Marco Antonio Peña-Duque, Marco Antonio Martínez-Ríos, Hilda Delgadillo-Rodríguez, José Manuel Fragoso

**Affiliations:** 1Department of Molecular Biology, Instituto Nacional de Cardiología Ignacio Chávez, Mexico City 14080, Mexico; gilberto.vargas@cardiologia.org.mx; 2Department of Endocrinology, Instituto Nacional de Cardiología Ignacio Chávez, Mexico City 14080, Mexico; julian.ramirez@cardiologia.org.mx; 3Department of Innovation and Technological Development, Instituto Nacional de Cardiología Ignacio Chávez, Mexico City 14080, Mexico; marcopduque@gmail.com; 4Department of Hemodynamics, Instituto Nacional de Cardiología Ignacio Chávez, Mexico City 14080, Mexico; marco.martinez@cardiologia.org.mx (M.A.M.-R.); hilda.delgadillo@cardiologia.org.mx (H.D.-R.)

**Keywords:** genetics, susceptibility, polymorphisms, restenosis

## Abstract

In the present study, we evaluated the association of the *BAT1*, *NFKBIL*, *LTA*, and *CASP1* single nucleotide polymorphisms and the gene–gene interactions with risk of developing restenosis after coronary stenting. The allele and genotype determination of the polymorphisms (*BAT1* rs2239527 *C*/*G*, *NFKBIL1* rs2071592 *T*/*A*, *LTA* rs1800683 *G*/*A*, *CASP1* rs501192 *A*/*G*, and *CASP1* rs580253 *A*/*G*) were performed by 5’exonuclease TaqMan assays in 219 patients: 66 patients with restenosis and 153 without restenosis. The distribution of rs2239527 *C*/*G*, rs2071592 *T*/*A*, and rs1800683 *G*/*A* polymorphisms was similar in patients with and without restenosis. Nonetheless, under recessive (OR = 2.73, pC_Res_ = 0.031) and additive models (OR = 1.65, pC_Add_ = 0.039), the *AA* genotype of the rs501192 *A*/*G* polymorphism increased the restenosis risk. Under co-dominant, dominant, recessive, and additive models, the *AA* genotype of the rs580253 A/G was associated with a high restenosis risk (OR = 5.38, pC_Co-Dom_ = 0.003; OR = 2.12, pC_Dom_ = 0.031; OR = 4.32, pC_Res_ = 0.001; and OR = 2.16, 95%CI: 1.33–3.52, pC_Add_ = 0.001, respectively). In addition, we identified an interaction associated with restenosis susceptibility: *BAT1-NFKBIL1-LTA-CASP1* (OR = 9.92, *p* < 0.001). In summary, our findings demonstrate that the rs501192 *A*/*G* and rs580253 *A*/*G* polymorphisms, as well as the gene–gene interactions between *BAT1-NFKBIL1-LTA-CASP1*, are associated with an increased restenosis risk after coronary stenting.

## 1. Introduction

At present, the treatment strategy for coronary artery disease (CAD) has been the intracoronary stent; after its placement, however, a percentage of the patients (12% to 32%) have developed restenosis [1,2,3,4]. This clinical entity is an arterial wall healing response to mechanical injury and comprises the following processes: vessel remodeling and neointimal hyperplasia, which involves smooth muscle migration/proliferation and extracellular matrix deposition [2,3,4]. 

Previous work has shown that the proteins encoded by the branched-chain amino-acid transaminase 1 (*BAT1*), NF-kB inhibitor-like-1 (*NFKBIL1*), lymphotoxin-alpha (LTA), and Caspase-1 (*CASP1*) genes play an important role in the regulation of inflammatory process and atherosclerotic plaque development [5,6,7,8]. *BAT1* encodes a nuclear protein called HLA-B-associated transcript 1, which downregulates the expression of pro-inflammatory cytokines such as interleukin-1 (IL-1), interleukin-6 (IL-6), and tumor necrosis factor (TNF) [9,10]. A similar effect has been reported for the protein that is homologous to kB inhibitors (IkB) and encoded by the *NFKBIL1* gene. This protein negatively regulates the expression of nuclear factor-kappa B transcription factor (NF-kB), which mediates the biosynthesis of pro-inflammatory cytokines [11]. LTA binds to the TNF receptors (TNFR-1 and -2) and regulates the inflammatory process and cell differentiation [12]. In addition, the *CASP1* gene has an important role in the maturation of the inactive pro-IL-1B and pro-IL-18 precursors of mature cytokines [13,14]. 

The *BAT1*, *NFKBIL1*, and *LTA* genes are located in the 6q21.3 region. Recent evidence has shown an association between three polymorphisms of these genes [*BAT1* rs2239527 C/G, *NFKBIL1* rs2071592 *T*/*A*, and *LTA* rs1800683 *A*/*G*] with the presence of acute coronary syndrome (ACS), myocardial infarction, and hypertension [5,6,7,8]. On the other hand, the *CASP1* gene is located on chromosome 11q23 and presents two relevant polymorphisms [rs501192 *A*/*G* and rs580253 *A*/*G*] associated with ACS [15]. However, little is known about the association of *CASP1* gene polymorphisms and restenosis after coronary stenting [16,17]. 

Considering the prominent role of the *BAT1*, *NFKBIL1*, *LTA*, and *CASP1* genes in the inflammatory process regulation, our aim was to establish an association of the rs2239527 *C*/*G*, rs2071592 *T*/*A*, rs1800683 *G*/*A*, rs501192 *A*/*G*, and rs580253 *A*/*G* polymorphisms with the susceptibility to restenosis after coronary stent placement. 

## 2. Material and Methods 

### 2.1. Study Population 

This cohort study was carried out at the Instituto Nacional de Cardiología Ignacio Chávez in Mexico. The sample size was calculated for a cohort study matched with a power of 80% and an alpha level of 0.05 (http://www.openepi.com/SampleSize/SSCohort.htm (accessed on 17 March 2021)). The total sample size required to carry out this study was 154 Mexican mestizo individuals (77 patients with restenosis and 77 patients without restenosis). Nonetheless, in our study, we included 219 patients with CAD who underwent coronary stent implantation at our institution between October 2008 and October 2014. After six months, these patients went to a follow-up coronary angiography; they presented with ischemia symptoms documented in a myocardial perfusion imaging test. Basal and procedure coronary angiographies were analyzed for angiographic predictors of restenosis, and a follow-up angiography was performed to screen for binary restenosis. To define restenosis, we used a >50% stenosis criterion at follow-up (50% reduction of the stenosis luminal diameter compared with the coronary angiography findings immediately after angioplasty). As a result, 66 patients were placed in the restenosis group and 153 patients in the non-restenosis group. As a comparison group, we included 625 healthy individuals with neither symptoms nor previous diagnosis of cardiovascular or systemic disease previously reported [5,15]. All subjects were considered Mexican mestizos and ethnically matched according to ancestry informative markers (AIMs) in the ADMIXTURE software. The study complied with the Declaration of Helsinki and was approved by the Ethics and Research commission of the Instituto Nacional de Cardiología Ignacio Chávez (number project: 22-1288). Written informed consent was obtained from all individuals enrolled in the study.

### 2.2. Laboratory Analysis

The plasma concentrations of the cholesterol and triglycerides were determined by enzymatic/colorimetric assays (Randox Laboratories, Crumlin, UK). The high-density lipoprotein-cholesterol (HDL-C) plasma concentrations were determined by the method of the phosphotungstic acid-Mg^2+^. The low-density lipoprotein-cholesterol (LDL-C) concentrations were determined in samples with a triglyceride level lower than 400 mg/dL with the Friedewald formula [18]. Dyslipidemia was defined as the presence of one or more of the following conditions: cholesterol > 200 mg/dL, LDL-C > 130 mg/dL, HDL-C < 40 mg/dL, or triglycerides > 150 mg/dL, according to the guidelines of the National Cholesterol Education Project (NCEP) Adult Treatment Panel (ATP III) (http://www.nhlbi.nih.gov/guidelines/cholesterol/atp3_rpt.htm, https://www.msdmanuals.com/professional/SearchResults?query=Normal+laboratory+values (accessed on 6 October 2021)). According to the MSD manual guidelines, type 2 diabetes mellitus (T2DM) was considered when participants had a fasting glucose level ≥ 126 mg/dL, previously diagnosed by a physician. In addition, hypertension was defined by a systolic blood pressure ≥ 140 mmHg, diastolic blood pressure ≥ 90 mmHg, or the use of oral antihypertensive therapy (https://www.msdmanuals.com/professional/SearchResults?query=Normal+laboratory+values (accessed on 6 October 2021)).

### 2.3. Genetic Analysis

According to the Lahiri and Nurnberger procedure, DNA extraction was performed from peripheral blood [19]. The allele and genotype determination of the *BAT1* rs2239527 *C*/*G*, *NFKBIL1* rs2071592 *T*/*A*, *LTA* rs1800683 *G*/*A*, *CASP1* rs501192 *A*/*G*, and *CASP1* rs580253 *A*/*G* polymorphisms were performed by 5’exonuclease TaqMan assays on a 7900HT Fast Real-Time PCR system accordance with manufacturer’s instructions (Applied Biosystems, Foster City, CA, USA) (Table 1). To avoid genotyping errors, 10% of the samples were determined twice, and all results were concordant.

### 2.4. Genetic Interactions Analysis

The genetic interactions between *BAT1* rs2239527 *C*/*G*, *NFKBIL1* rs2071592 *T*/*A*, *LTA* rs1800683 *G*/*A*, *CASP1* rs501192 *A*/*G*, and *CASP1* rs580253 *A*/*G* polymorphisms in patients with and without restenosis were performed by the multifactor dimensionality reduction (MDR) program. This program is useful for the study of complex diseases whose mode of genetic inheritance is unknown. In addition, it evaluates statistical epistasis, i.e., non-additive gene–gene interactions between genotypes of different loci, as well as the quantitative differences of allele-specific effects, randomly validating the internal data through training, testing, and validation analysis. The interactions between multilocus genotypes are estimated using *p* values: values less than 0.05 mean that they are associated with the disease [20]. 

### 2.5. Analysis of the Haplotypes

Haplotype design and linkage disequilibrium analysis (LD, D’) were performed using Haploview version 4.1 (Broad Institute of Massachusetts Institute of Technology and Harvard University, Cambridge, MA, USA). This software analyzes the combination of alleles in a single gene, or alleles in multiple genes along a chromosome, that tend to be inherited together due to the proximity between them, providing the statistical calculation of the linkage disequilibrium (LD, D’), LOD and r squared, as well as possible haplotype patterns from primary genotypes, using the platform of the Human Haplotype Map project data [21].

### 2.6. Statistical Analysis

Gene frequencies of *BAT1*, *NFKBIL*, *LTA*, and *CASP1* polymorphisms in patients with and without restenosis were obtained by direct counting. The Hardy–Weinberg equilibrium was evaluated by the chi-squared test. The data analysis was performed with SPSS version 18.0 (SPSS, Chicago, IL, USA). Either the Mann–Whitney *U* or the Student’s *t*-test was used to compare the continuous variables of the study groups. For categorical variables, a chi-squared or Fisher’s exact test was performed. The associations of the polymorphisms with restenosis were evaluated by logistic regression analysis under the following inheritance model: additive, co-dominant, dominant, over-dominant, and recessive, adjusting according to cardiovascular risk factors. All *p*-values were corrected (pC) by the Bonferroni test. The pC values < 0.05 were considered statistically significant, and all odds ratios (OR) were estimated with 95% confidence intervals. In this study, the restenosis occurrence was based on the following OR cases: (a) OR = 1 does not affect the odds of developing restenosis, (b) OR > 1 is associated with higher restenosis odds, and (c) OR < 1 is associated with lower odds. The statistical power to detect an association of restenosis was 0.80 according to the OpenEpi software (http://www.openepi.com/SampleSize/SSCC.html (accessed on 17 March 2021)). 

## 3. Results

### 3.1. Characteristics of the Study Population

The clinical and angiographic characteristics of the patients with and without restenosis are presented in Table 2. As can be seen, despite statin therapy, the levels of total cholesterol and HDL-C in patients with restenosis were significantly different (*p*-value < 0.05) from those in patients without restenosis. In addition, restenosis was more frequent in patients who underwent coronary bare-metal stent (BSM) implantation (73%) than in those with drug-eluting stent (DES) implantation (27%) (*p* ≤ 0.001). The presence of stable angina was less common in patients with restenosis (9%) than in those without restenosis (22%). On the other hand, unstable angina and a stent diameter less than or equal to 2.5 mm were more frequent in patients with restenosis (39%, and 30%, respectively) than in those without this clinical entity (28% and 20%, respectively). 

#### Association of CASP1 Polymorphisms with Restenosis

Genotype frequencies in the polymorphic sites were in Hardy–Weinberg equilibrium. In a first analysis, the genetic distribution of the rs2239527 *C*/*G*, rs2071592 *T*/*A*, and rs1800683 *G*/*A* polymorphisms were similar in patients with CAD and healthy individuals. Nonetheless, the genotype frequencies of the rs580253 *A*/*G* and rs501192 *A*/*G* SNPs showed significant differences between CAD patients and healthy individuals (*p* < 0.05) (Appendix A). In addition, when we realized the analysis of the rs580253 *A*/*G* and rs501192 *A*/*G* polymorphisms comparing patients with and without restenosis, we showed that under recessive and additive models, the *AA* genotype of the rs501192 *A*/*G* polymorphism increased the risk of developing restenosis (OR = 2.73, 95%CI: 1.10–6.81, pC_Res_ = 0.031 and OR = 1.65, 95%CI: 1.02–2.65, pC_Add_ = 0.039, respectively). In addition, under co-dominant, dominant, recessive, and additive models, the *AA* genotype of rs580253 *A*/*G* was associated with a high restenosis risk (OR = 5.38, 95%CI: 1.99–14.6, pC_Co-Dom_ = 0.003; OR = 2.12, 95%CI: 1.06–4.23, pC_Dom_ = 0.031; OR = 4.32, 95%CI: 1.76–10.6, pC_Res_ = 0.001; and OR = 2.16, 95%CI: 1.33–3.52, pC_Add_ = 0.001, respectively) (Table 3). All models were adjusted for age, gender, blood pressure, BMI, glucose, total cholesterol, HDL-C, LDL-C, triglycerides, hypertension, T2DM, dyslipidemia, and smoking habit, as well as with the angiographic characteristics, such as unstable angina, stable angina, drug-eluting stent (DES), bare metal stent (BMS), diameter smaller, and stent length.

### 3.2. Linkage Disequilibrium Analysis

The rs2239527 *C*/*G*, rs2071592 *T*/*A*, and rs1800683 *G*/*A* polymorphisms showed a strong linkage disequilibrium (D’ > 0.95). Our analysis revealed three (*CTG*, *GAA*, and *CTA)* haplotypes without significant differences between patients with and without restenosis (Table 4). On the other hand, the analysis of the rs580253 *A*/*G* and rs501192 *A*/*G* SNPs showed a linkage disequilibrium (D’ > 0.85). This analysis exhibited three (*GG*, *GA*, and *AG*) out of four haplotypes with significant differences between the two groups (Table 4). The “*GA*” and “*AG*” haplotypes proved to be more common in patients with restenosis (11.4% and 6.9%, respectively) than in those without the condition (0.1% and 0.4%, respectively). In contrast, the “GG” haplotype was more frequent in the non-restenosis group (68.9%) than in the restenosis group (46.1%) (Table 4). 

### 3.3. Interaction Analysis

The interaction analysis showed that three out of four interactions between the *BAT1*, *NFKBIL1*, *LTA*, and *CASP1* genotypes and restenosis were statistically significant (Table 5). The best model of interaction associated with risk of the developing restenosis was formed by rs2239527 *C*/*G*, rs2071592 *T*/*A*, rs1800683 *G*/*A*, rs501192 *A*/*G*, and rs580253 *A*/*G* (training accuracy = 0.722, testing accuracy = 0.683, cross-validation consistency = 10/10, OR = 9.92 and *p* < 0.001). In addition, it may be seen in Table 5 that there are three other interactions associated with risk of the developing restenosis: rs2239527 *C*/*G*, rs2071592 *T*/*A*, rs501192 *A*/*G*, and rs580253 *A*/*G* (training accuracy = 0.710, testing accuracy = 0.668, cross-validation consistency = 8/10, OR = 9.03, and *p* < 0.001), rs2071592 *T*/*A*, rs501192 *A*/*G*, and rs580253 A/G (training accuracy = 0.695, testing accuracy = 0.651, cross-validation consistency = 9/10, OR = 8.00, and *p* < 0.001), and rs501192 *A*/*G*, and rs580253 A/G (training accuracy = 0.657, testing accuracy = 0.657, cross-validation consistency = 10/10, OR = 5.53, and *p* < 0.001).

## 4. Discussion

Recent studies have shown that the polymorphisms (*BAT1* rs2239527 *C*/*G*, *NFKBIL1* rs2071592 *T*/*A*, *LTA* rs1800683 *G*/*A*, *CASP1* rs501192 *A*/*G*, and *CASP1* rs580253 *A*/*G*) are associated with the presence of ACS, myocardial infarction, and CAD [5,6,7,8,15]. In this study, we found that the rs580253 *A* and rs501192 *A* alleles of the *CASP1* gene, as well as the gene–gene interaction between *BAT1*, *NFKBIL1*, *LTA*, and *CASP1*, were associated with an increased risk of developing restenosis after coronary stenting. As far as we know, our work is one of few studies that describe the association of the rs580253 *A*/*G* and rs501192 *A*/*G* polymorphisms with restenosis risk after coronary stenting. In line with our data, Bergheanu et al. reported that carriers of the *A* allele of the rs580253 *A*/*G* polymorphism were at greater risk for restenosis after coronary stenting in a Caucasian population [17]. Similar data were reported by Monraats et al., who studied the same polymorphism (rs580253 *A*/*G*) in patients with restenosis; in this study, the authors described that the *AA* genotype is associated with the risk of developing restenosis in a Caucasian population [16]. In addition, we reported that the *AA* genotype of the rs501192 *A*/*G* polymorphism increased the risk of developing restenosis. In contrast with this data, Blankenberg et al. reported that in Caucasian patients with CAD, the *A* allele of the rs501192 *A*/*G* polymorphism exhibited a borderline association with a decreased risk of developing CAD (*p* = 0.053) [22]. Alternatively, prior studies have shown (with positive and negative results in different populations) that the rs580253 *A*/*G* and rs501192 *A*/*G* polymorphisms have an important role in the development of diseases such as ACS, Chagas cardiomyopathy, Alzheimer’s, and cancer [15,23,24,25]. On the other hand, our results showed an increased frequency of the haplotypes (*AG* and *GA*) conformed by rs580253 *A*/*G* and rs501192 *A*/*G* polymorphisms in patients with restenosis. In this context, we suggested that these haplotypes are associated with the risk of developing restenosis after coronary stenting due to both haplotypes, including the rs580253 *A* and rs501192 *A* alleles, which were associated independently with the disease. Therefore, this finding corroborates the role of these two alleles (analyzed either independently or as haplotypes) in the genetic susceptibility to restenosis. On the other hand, the information about these polymorphisms is scarce in other populations. We think that the association of the *CASP1* polymorphisms with restenosis may also be due to the allelic distribution of these polymorphisms, which varies according to the ethnic origin of the study populations. In this context, we reported that the frequency of the rs501192 *A* and rs580253 *A* alleles in the Mexican population is 32.4% and 32.5%, respectively (Appendix A). Similar data showed individuals from Los Angeles with Mexican ancestry (30%, and 30%, respectively), according to data obtained from NCBI (National Center for Biotechnology Information) (https://www.ensembl.org/index.html (accessed on 17 September 2021)). Nonetheless, the allelic distribution of these polymorphisms was less frequent in Caucasian (19% and 19%, respectively), African (9% and 9%, respectively), and Asian (1% and 1%, respectively) populations, according to data obtained from NCBI (National Center for Biotechnology Information) (https://www.ensembl.org/index.html (accessed on 17 September 2021)). Considering our results and the distribution of the alleles and genotypes of the rs501192 *A*/*G* and rs580253 *A*/*G* polymorphisms in our population, we propose that additional studies in other populations with different ethnic origins could help define the true role of these polymorphisms as markers of risk in the developing restenosis after coronary stenting. 

In the last decade, several studies have shown that the analysis of gene–gene interactions between different genes plays an important role in the risk of developing complex diseases such as coronary heart disease, rheumatoid arthritis, and hyperlipidemia [26,27,28,29]. In this study, we evaluated whether the gene–gene interaction between *BAT1* rs2239527 *C*/*G*, *NFKBIL1* rs2071592 *T*/*A*, *LTA* rs1800683 *G*/*A*, *CASP1* rs580253 *A*/*G,* and *CASP1* rs501192 *A*/*G* polymorphisms could be jointly associated with the risk of developing restenosis after stenting using the multifactor dimensionality reduction (MDR) program. MDR evaluates statistical epistasis based on models of genetic interaction between genotypes of different loci [20]. The MDR is a mathematical model based on algorithms that can make predictions. This program implements the training set for all possible combinations of loci, and the x models with the highest balanced accuracy are retained for evaluation in the testing set. MDR performs on all x models in the testing set, and the best model for each level of interaction is preserved for evaluating predictive ability in the validation set. Finally, one model is chosen to maximize balanced accuracy in the validation set [30]. Interestingly, this analysis showed four interaction models associated with an increased risk of developing restenosis after stenting: (I) the model formed by rs501192 *A*/*G* and rs580253 *A*/*G* (OR = 5.53); (II) the model composed by rs2239527 *C*/*G*, rs2071592 *T*/*A*, rs1800683 *G*/*A*, rs501192 *A*/*G*, and rs580253 *A*/*G* (OR = 9.92); (III) model that consists of rs2239527 *C*/*G*, rs2071592 *T*/*A*, rs501192 *A*/*G*, and rs580253 *A*/*G* (OR = 9.02); and (IV) model composed by rs2071592 *T*/*A*, rs501192 *A*/*G*, and rs580253 *A*/*G* (OR = 8.00). However, it is important to mention that the polymorphisms rs2239527 *C*/*G*, rs2071592 *T*/*A*, and rs1800683 *G*/*A* individually showed no evidence of association with restenosis after coronary stenting. Nonetheless, when we analyzed the gene–gene interactions between the five SNPs of the four genes, interestingly, several of them showed a strong association with this pathology, increasing the OR values dramatically. In this context, the best model of gene–gene interaction was between the rs2239527 *C*/*G*, rs2071592 *T*/*A*, rs1800683 *G*/*A*, rs501192 *A*/*G*, and rs580253 *A*/*G* polymorphisms, in accordance with OR value and cross-validation consistency. As far as we know, this is the first report that showed an association of the gene–gene interaction between the *BAT1*, *NFKBIL1*, *LTA*, and *CASP1* SNPs with a higher risk of the development of restenosis after coronary stenting in our population. Nonetheless, we considered that these findings should be taken with caution. Because the gene–gene interactions are only statistically associated between different genes identified by the MDR program, it is not known whether these gene–gene interactions could play a biologically relevant role. Considering our data, future investigations are warranted to understand the contribution of the gene–gene interactions between the *BAT1*, *NFKBIL1*, *LTA*, and *CASP1* SNPs and the risk of developing restenosis after coronary stenting or some other cardiovascular event. 

In summary, we found that the rs580253 *A*/*G* and rs501192 *A*/*G* polymorphisms of the *CASP1* gene, as well as the gene–gene interactions between *BAT1-NFKBIL1-LTA-CASP1*, are associated with the risk of developing restenosis after coronary stenting. Furthermore, we determined that two haplotypes (*AG* and *GA*) composed of the rs580253 *A*/*G* and rs501192 *A*/*G* polymorphisms were associated with the highest risk of developing restenosis after coronary stenting. Finally, based on our results and considering the specific genetic characteristics of the Mexican population, we propose that additional studies in other populations with different ethnic origins could help define the true role of these polymorphisms as markers of risk in the developing restenosis after coronary stenting. 

## Figures and Tables

**Table 1 biomolecules-12-00765-t001:** Information on the gene polymorphism tested.

Gene Symbol	dbSNP ^a,b^	Chromosome	Chromosome Position	Change Base (pb)	Location in Gene
*BAT1*	rs2239527	6p21.3	31542002	*C > G*	NearGene-5
*NFKBIL1*	rs2071592	6p21.3	31547563	*T > A*	NearGene-5
*LTA*	rs1800683	6p21.3	31572294	*A > G*	5′-untranslated region
*CASP1*	rs501192	11q23	105029658	*A > G*	Intron variant
*CASP1*	rs580253	11q23	105029761	*A > G*	Coding Sequence variant

^a^ SNP ID in database dbSNP; ^b^ Given name according to NCBI.

**Table 2 biomolecules-12-00765-t002:** Baseline clinical and angiographic characteristics of the CAD patients (patients with and without restenosis).

Clinical Characteristics		With Restenosis (*n* = 66)	Without Restenosis (*n* = 153)	*p*-Value
Age (years)		60.4 ± 10.88	59.12 ± 10.49	0.657
BMI (kg/m^2^)		26.6 (24.1–29.7)	26.1 (24.2–29)	0.601
Blood pressure (mmHg)	Systolic	120 (110–140)	120 (110–130)	0.254
	Diastolic	80 (70–84)	80 (70–80)	0.184
Glucose (mg/dL)		120 (91–153)	116 (95–159)	0.312
Total cholesterol (mg/dL)		150 (113–181)	166 (133–205)	0.004
HDL-C (mg/dL)		41 (36–48)	38 (33–49)	0.009
LDL-C (mg/dL)		102 (63–131)	102 (70–138)	0.657
Triglycerides (mg/dL)		150 (109–207)	169 (126–215)	0.159
Gender *n* (%)	Male	51 (77)	119 (78)	0.274
	Female	15 (23)	34 (22)	
Hypertension *n* (%)	Yes	32 (48)	66 (43)	0.232
Type II diabetes mellitus *n* (%)	Yes	33 (50)	72 (47)	0.344
Dyslipidemia *n* (%)	Yes	50 (76)	127 (83)	0.144
Smoking *n* (%)	Yes	41 (62)	93 (61)	0.463
Unstable angina *n* (%)	Yes	26 (39)	43 (28)	0.042
Stable angina *n* (%)	Yes	6 (9)	34 (22)	0.017
Statin therapy *n* (%)	Yes	53 (80)	131 (85)	0.216
DES *n* (%)	Yes	18 (27)	94 (61)	<0.0001
BSM *n* (%)	Yes	48 (73)	59 (38)	<0.0001
Stent Diameter ≤ 2.5 mm *n* (%)	Yes	20 (30)	30 (20)	0.041
Stent length < 20 mm *n* (%)	Yes	39 (59)	85 (55)	0.368

Data are expressed as median and percentiles (25th–75th). *p*-values were estimated using Mann–Whitney *U* test for continuous variables and chi-square test for categorical values. Abbreviations: BMS, bare metal stent; DES, drug-eluting stent.

**Table 3 biomolecules-12-00765-t003:** The inheritance models analysis of *CASP1* polymorphisms in patients with and without restenosis.

SNP	Model	Genotype	With Restenosis *n* = 66 *n* (%)	Without Restenosis *n* = 219 *n* (%)	OR (95%CI)	*pC*
rs501192	Co-dominant	*GG*	25 (0.379)	76 (0.497)	3.11 (1.15–8.40)	0.078
		*GA*	26 (0.386)	59 (0.386)		
		*AA*	15 (0.227)	18 (0.118)		
	Dominant	*GG*	25 (0.379)	76 (0.497)	1.59 (0.81–3.13)	0.169
		*GA + AA*	41 (0.621)	77 (0.503)		
	Recessive	*GG + GA*	51 (0.773)	135 (0.882)	2.73 (1.10–6.81)	0.031
		*AA*	15 (0.227)	18 (0.118)		
	Over-dominant	*GG + AA*	40 (0.606)	94 (0.614)	0.93 (0.48–1.81)	0.839
		*GA*	26 (0.394)	59 (0.386)		
	Log-additive	*--------*	---------	-----------	1.65 (1.02–2.65)	0.039
rs580253	Co-dominant	*GG*	22 (0.333)	77 (0.503)	5.38 (1.99–14.6)	0.003
		*GA*	26 (0.394)	58 (0.379)		
		*AA*	18 (0.273)	18 (0.118)		
	Dominant	*GG*	22 (0.333)	77 (0.503)	2.12 (1.06–4.23)	0.031
		*GA + AA*	44 (0.667)	76 (0.497)		
	Recessive	*GG + GA*	48 (0.727)	135 (0.882)	4.32 (1.76–10.6)	0.001
		*AA*	18 (0.273)	18 (0.118)		
	Over-dominant	*GG + AA*	40 (0.606)	95 (0.621)	0.91 (0.46–1.76)	0.769
		*GA*	26 (0.394)	58 (0.379)		
	Log-additive	*--------*	---------	-----------	2.16 (1.32–3.52)	0.001

Abbreviations: OR, odds ratio; CI, confidence interval; MAF, minor allele frequency; pC, *p*-value. The *p*-values were calculated by the logistic regression analysis. ORs were adjusted for age, gender, blood pressure, BMI, glucose, total cholesterol, HDL-C, LDL-C, triglycerides, hypertension, T2DM, dyslipidemia, and smoking. We also adjusted these ratios for the angiographic characteristics, such as unstable angina, stable angina, drug-eluting stent (DES), bare metal stent (BMS), diameter smaller, and stent length.

**Table 4 biomolecules-12-00765-t004:** Distribution of the haplotypes between the *BAT1* rs2239527 *C*/*G*, *NFKBIL1* rs2071592 *T*/*A*, and *LTA* 1800683 *A*/*T* polymorphisms, and the *CASP1* rs501192 *A*/*G* and *CASP1* rs580253 *A*/*G* polymorphisms in the study groups.

SNP	With Restenosis *n* = 66	Without Restenosis *n* = 153	
rs2239527–rs2071592–rs1800683	Hf (%)	Hf (%)	*p*
*CTG*	0.628	0.608	0.759
*GAA*	0.311	0.366	0.316
*CTA*	0.023	0.010	0.535
rs501192–rs580253	Hf (%)	Hf (%)	*p*
*GG*	0.461	0.689	1 × 10^−4^
*AA*	0.355	0.306	0.371
*GA*	0.114	0.001	<0.001
*AG*	0.069	0.004	1 × 10^−3^

Abbreviations: Hf, haplotype frequency; *p* = *p*-value. The polymorphism order in the different haplotypes is according to the positions in the chromosome. (rs2239527–rs2071592–rs1800683 chromosome 6q21.3, and rs501192-rs580523-chromosome 11q23).

**Table 5 biomolecules-12-00765-t005:** Gene–gene interactions between *BAT1* rs2239527 *C*/*G*, *NFKBIL1* rs2071592 *T*/*A*, *LTA* rs1800683 *G*/*A*, *CASP1* rs501192 *A*/*G*, and *CASP1* rs580253 *A*/*G* polymorphisms in patients with restenosis vs. without restenosis.

Interactions Associated	Training Accuracy	Testing Accuracy	CVC **	*p*-Value	OR (CI 95%)
rs501192 *A*/*G*, and rs580253 *A*/*G*	0.657	0.657	10/10	< 0.001	5.53 (2.79–10.95)
rs2071592 *T*/*A*, rs501192 *A*/*G*, and rs580253 *A*/*G*	0.695	0.651	9/10	< 0.001	8.00 (3.98–16.01)
rs2239527 *C*/*G*, rs2071592 *T*/*A*, rs501192 *A*/*G*, and rs580253 *A*/*G*	0.710	0.668	8/10	< 0.001	9.03 (4.49–18.16)
rs2239527 *C*/*G*, rs2071592 *T*/*A*, rs1800683 *G*/*A*, rs501192 *A*/*G*, and rs580253 *A*/*G*	0.722	0.683	10/10	< 0.001	9.92 (4.75–20.7) *

* The best model is referred to as the one with the maximum testing accuracy and maximum CVC ** (cross-validation consistency).

## Data Availability

The data presented in this study are available upon request from the corresponding author.

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
