# Peer review of "CASP1 Gene Polymorphisms and BAT1-NFKBIL-LTA-CASP1 Gene–Gene Interactions Are Associated with Restenosis after Coronary Stenting"

_biomolecules, 2022, doi:10.3390/biom12060765_

Round 1
Reviewer 1 Report
The reviewed paper is devoted to an important and relevant scientific problem covering the search for associations of certain genetic factors with an increased risk of restenosis in patients who have undergone coronary artery stenting. In particular, the polymorphism of genes CASP1 and BAT1-NFKBIL-LTA-CASP1 gene-gene interactions was studied. Data on genetic predisposition to the development of coronary artery restenosis can serve as an important prognostic marker in patients who are scheduled for stenting in case of coronary artery disease. The study included 219 patients with CAD who underwent coronary artery stenting. The total sample size was 154 Mexican metizo persons: 77 with restenosis and 77 – without restenosis. Laboratory and genetic studies were carried out. Using the method of real-time PCR polymorphism of the BAT1 - 23C/G (rs2239527), NFKBIL1_-63T/A (rs2071592), LTA -162G/A (rs1800683), CASP1 G+7/in6A (rs501192), and A10370-G Exon 6 (rs580253) was performed. The genetic interactions between BAT1, NFKBIL, LTA, and CASP1 polymorphisms were also performed by the multifactor dimensionality reduction (MDR) program in patients with and without restenosis. It was found that CASP1 G+7/in6A and A10370-G polymorphisms, as well as the interactions gene-gene between BAT1-NFKBIL1-LTA-CASP1 are associated with a risk of restenosis. The paper would seem quite interesting and useful, however, the following remarks require a major revision:
- There is no comparison group of individuals without coronary artery disease. At least it is necessary to take data with control indicators for the frequencies of genes and genotypes from other publications on this ethnic group or, as a last resort, to compare with population frequencies (but it is not very desirable).
- It is not very correct to analyze data for patients with bare-metal and drug-eluting stents together as bare-metal stents generally pose a higher risk of restenosis.
- In the discussion or in a separate table, a brief description of each polymorphism should be given: in particular, how do different polymorphic variants affect the function of the corresponding gene, as well as what are the possible mechanisms for their involvement in the development of restenosis.
- Quite a lot of old literature sources are cited.
Author Response
Thank you very much for your comments and suggestions to Manuscript ID biomolecules-1720661
We would like to thank the Reviewer for their comments; they have helped to improve the manuscript
Comments to the Author:
Reviewer #1: Comments and Suggestions for Authors
The reviewed paper is devoted to an important and relevant scientific problem covering the search for associations of certain genetic factors with an increased risk of restenosis in patients who have undergone coronary artery stenting. In particular, the polymorphism of genes CASP1 and BAT1-NFKBIL-LTA-CASP1 gene-gene interactions were studied. Data on genetic predisposition to the development of coronary artery restenosis can serve as an important prognostic marker in patients who are scheduled for stenting in case of coronary artery disease. The study included 219 patients with CAD who underwent coronary artery stenting. The total sample size was 154 Mexican metizo persons: 77 with restenosis and 77 – without restenosis. Laboratory and genetic studies were carried out. Using the method of real-time PCR polymorphism of the BAT1 - 23C/G (rs2239527), NFKBIL1_-63T/A (rs2071592), LTA -162G/A (rs1800683), CASP1 G+7/in6A (rs501192), and A10370-G Exon 6 (rs580253) was performed. The genetic interactions between BAT1, NFKBIL, LTA, and CASP1 polymorphisms were also performed by the multifactor dimensionality reduction (MDR) program in patients with and without restenosis. It was found that CASP1 G+7/in6A and A10370-G polymorphisms, as well as the interactions gene-gene between BAT1-NFKBIL1-LTA-CASP1 are associated with a risk of restenosis. The paper would seem quite interesting and useful, however, the following remarks require a major revision:
Answer: Thanks for the comments, however, we clarified that the sample size required to carry out this study is of 154 Mexican mestizo individuals (77 patients with restenosis and 77 patients without restenosis) calculated by http://www.openepi.com/SampleSize/SSCohort.htm. Nonetheless, in our study, we included 219 patients with CAD, who underwent coronary stent implantation, 66 patients development restenosis, and 153 patients not development restenosis.
- There is no comparison group of individuals without coronary artery disease. At least it is necessary to take data with control indicators for the frequencies of genes and genotypes from other publications on this ethnic group or, as a last resort, to compare with population frequencies (but it is not very desirable).
Answer: In response to comment of the reviewer. Unfortunately, we not included individuals without CAD in this study. However, we previously reported the allele and genotype frequencies of the polymorphisms [BAT1 -23C/G (rs2239527), NFKBIL1_-63T/A (rs2071592), LTA -162G/A (rs1800683), CASP1 G+7/in6A (rs501192), and A10370-G (rs580253)] in approximately 617 healthy individuals Mexican (Immunol Res. 2017; 65: 862-868, and Rev Invest Clin. 2020; 72: 19-24). We think that these data could be serve as a reference to make the comparison between CAD patients and healthy controls.
In this context, we included in a supplementary Table, the allele and genotype frequencies of these SNPs between CAD patients and healthy controls.
Table 2. Allele and genotype frequencies of BAT1 rs2239527 C/G, NFKBIL1 rs2071592 T/A, LTA rs1800683 G/A, CASP1 rs501192 A/G, and CASP1 rs580253 A/G polymorphisms in CAD patients and healthy controls
|
Polymorphic site (rsID-number) |
CAD n=219 (n(%)) |
Controls n=617 (n(%)) |
*p |
|
rs2239527 C/G |
|
|
|
|
Allele |
|
|
|
|
C |
278 (63.4) |
810 (65.6) |
NS |
|
G |
160 (36.5) |
424 (34.3) |
|
|
Genotype |
|
|
|
|
CC |
87 (39.7) |
272 (44.1) |
|
|
CG |
104 (47.4) |
266 (43.1) |
NS |
|
GG |
28 (12.8) |
79 (12.8) |
|
|
rs2071592 T/A |
|
|
|
|
Allele |
|
|
|
|
T |
281 (64.1) |
811 (65.7) |
NS |
|
A |
157 (35.8) |
423 (34.2) |
|
|
Genotype |
|
|
|
|
TT |
89 (40.6) |
268 (43.4) |
|
|
TA |
103 (47.0) |
275 (44.6) |
NS |
|
AA |
27 (12.3) |
74 (12.0) |
|
|
rs1800683 G/A |
|
|
|
|
Allele |
|
|
|
|
G |
275 (62.7) |
794 (64.3) |
NS |
|
A |
163 (37.2) |
440 (35.6) |
|
|
Genotype |
|
|
|
|
GG |
86 (39.2) |
260 (42.1) |
|
|
GA |
103 (47.0) |
274 (44.4) |
NS |
|
AA |
30 (13.7) |
83 (13.5) |
|
|
rs501192 G/A |
|
|
|
|
Allele |
|
|
|
|
G |
287 (65.5) |
823 (67.5) |
NS |
|
A |
151 (34.4) |
395 (32.4) |
|
|
Genotype |
|
|
|
|
GG |
101 (46.1) |
277 (45.5) |
|
|
GA |
85 (38.8) |
269 (44.2) |
0.040 |
|
AA |
33 (15.0) |
63 (10.3) |
|
|
rs580253 A/G |
|
|
|
|
Allele |
|
|
|
|
G |
282 (64.4) |
821 (67.4) |
NS |
|
A |
156 (35.6) |
397 (32.5) |
|
|
Genotype |
|
|
|
|
GG |
99 (45.2) |
275 (45.2) |
|
|
GA |
84 (38.3) |
271 (44.5) |
0.011 |
|
AA |
36 (16.4) |
63 (10.3) |
|
Data are shown as n and frequency. *chi-square test.
In addition, we included the following phrase: “As a comparison group, we included 625 healthy individuals with neither symptoms nor previous diagnosis of cardiovascular or systemic disease previously reported (Immunol Res. 2017; 65: 862-868, and Rev Invest Clin. 2020; 72: 19-24)”. In material and methods section.
On the other hand, we added the following phrase: “Genotype frequencies in the polymorphic sites were in Hardy-Weinberg equilibrium. In a first analysis, the genetic distribution of the rs2239527 C/G, rs2071592 T/A, and rs1800683 G/A, polymorphisms were similar in patients with CAD and healthy individuals. Nonetheless, the genotype frequencies of the rs580253 A/G and rs501192 A/G SNPs showed significant differences between CAD patients and healthy individuals (p < 0.05) (supplementary Table)”. In result section.
In addition, we replaced the following phrase: “Under recessive and additive models,…” By “In addition, when we realized the analysis of the rs580253 A/G and rs501192 A/G polymorphisms comparing patients with and without restenosis, we showed that under recessive and additive models, …” In result section.
On the other hand, we replaced the following phrase: “In addition, the association of these polymorphisms with cardiovascular and other diseases in different populations is scarce and controversial. For example,…” BY “In line with our data,…” In discussion section.
In addition, we changed the following phrase: “We consider that further studies with a greater number of individuals and different ethnic origins are needed to explain the true role of CASP1 polymorphisms in the risk of developing restenosis.” BY “Considering our data and the distribution of the alleles and genotypes of the rs501192 A/G, and rs580253 A/G polymorphisms in our population, we propose that additional studies in other populations with different ethnic origin could help define the true role of these polymorphisms as markers of risk in the developing restenosis after coronary stenting.” In discussion section.
- It is not very correct to analyze data for patients with bare-metal and drug-eluting stents together as bare-metal stents generally pose a higher risk of restenosis.
Answer: In response to comment of the reviewer. We agree with this observation. However, the objective this work was not evaluated effect of the bare-metal and drug-eluting stents in the development of restenosis. Because data in the literature have been showed that both (bare-metal and drug-eluting stents) increased risk of development restenosis.
Nonetheless, when we analyzed our data, adjusting the inheritance models by angiographic and cardiovascular risk factors, using logistic regression analysis that allows estimating the relationship between the dependent variable and a set of independent variables. This analysis predicts whether the independent variables such as age, gender, blood pressure, BMI, glucose, total cholesterol, HDL-C, LDL-C, triglycerides, hypertension, T2DM, dyslipidemia, smoking habit. as well as with the angiographic characteristics, such as unstable angina, stable angina, drug eluting stent (DES), bare metal stent (BMS), diameter smaller, and stent length influence the risk of developing restenosis.
In this context, we mention that “All models were adjusted for age, gender, blood pressure, BMI, glucose, total cholesterol, HDL-C, LDL-C, triglycerides, hypertension, T2DM, dyslipidemia, smoking habit. as well as with the angiographic characteristics, such as unstable angina, stable angina, drug eluting stent (DES), bare metal stent (BMS), diameter smaller, and stent length.” In results section
- In the discussion or in a separate table, a brief description of each polymorphism should be given: in particular, how do different polymorphic variants affect the function of the corresponding gene, as well as what are the possible mechanisms for their involvement in the development of restenosis.
Answer: Agree with comment of the reviewer. We included as Table 1, a brief description of all polymorphisms studied in this work.
Table 1. Information of the gene polymorphism tested
|
Gene symbol |
dbSNP a,b |
Chromosome |
Chromosome position |
Change base (pb) |
Location in gene |
|
BAT1 |
rs2239527 |
6p21.3 |
31542002 |
C > G |
NearGene-5 |
|
NFKBIL1 |
rs2071592 |
6p21.3 |
31547563 |
T > A |
NearGene-5 |
|
LTA |
rs1800683 |
6p21.3 |
31572294 |
G > A |
5’-unstraslated region |
|
CASP1 |
rs501192 |
11q23 |
105029658 |
A > G |
Intron variant |
|
CASP1 |
rs580253 |
11q23 |
105029761 |
A > G |
Coding sequence variant |
a SNP ID in database dbSNP.
b Given name according to NCBI.
- Quite a lot of old literature sources are cited.
Answer: We agree with comment of the reviewer. We updated most of the references in the new version of the manuscript. However, not all the references could be refreshed, because many are necessary as scientific support for this work.
Reviewer 2 Report
Vargas-Alarcón et al. report that CASP1 gene polymorphisms and BAT1-NFKBIL-LTA-CASP1 gene-gene interactions are associated with restenosis after coronary stenting.
Major comments
Methods
Genetic analysis
- In the era of data/driven research, why and how did the Authors choose to pick a small number of polymorphisms, instead of performing whole genome/exome sequencing analyses of the patients?
Genetic interactions analysis
- More details on this analysis would be needed. How is a gene-gene interaction (epistasis?) being defined? Are these estimates based on 5 polymorphisms only? How is the background effect (i.e. when no gene-gene interactions exist) being estimated?
Analysis of the haplotypes
- More details on this analysis would be needed. Did the Authors estimate haplotypes based on 5 polymorphisms only? What reference population was being used? Were there any quality scores attached to these estimates?
Results
- Table 2 should be re-made by re-arranging cells in a different way, e.g. with vs. restenosis should be the columns, each of which could be further subdivided into the genetic models tested; genetic variants should be presented as rows, without empty cells in between, i.e. Overall brought closer to the standard representation of SNPs and their associations in the fields, e.g. Table 2 here https://www.frontiersin.org/articles/10.3389/fphar.2021.586973/full
- Table3 is currently also very hard to understand, each haplotype should represent one row for clarity, and empty cells should be avoided
- Interaction analysis and Table 4. This part is not clear at all. What does the number of factors represent? What do training and testing accuracy stand for? How was it calculated? How and with which data was the training performed?
Discussion
- „In the present study, we analyzed four genes and five polymorphisms“ Why only four genes and five polymorphisms, and exactly these genes and polymorphisms?
- Why did you perform gene-gene interactions, i.e. Why did you assume that the 5 SNPs in 4 genes should be „interacting“? What is the biological rationale beyond that?
- What is the medical utility of the 5 SNPs in 4 genes and the „interactions“ between them?
- „The association of these polymorphisms with cardiovascular and other diseases in different populations is scarce and controversial.“ Indeed, as the Authors state „additional studies in a larger number of individuals have to be undertaken.“ Would it be possible to get access to an independent validation cohort?
Minor comments
- Please, remove „.“ from the title and the listing of authors
Abstract
- „Five gene polymorphisms (BAT1 -23C/G, NFKBIL1_-63T/A, LTA -162G/A, CASP1 G+7/in6A, and A10370-G)“ please use the rsID numbers of the respective SNVs throughout the manuscript
- „The interactions gene-gene between BAT1-NFKBIL1-LTA-CASP1“ correct to „gene-gene interactions“
Author Response
Thank you very much for your comments and suggestions to Manuscript ID biomolecules-1720661
We would like to thank the Reviewer for their comments; they have helped to improve the manuscript
Comments to the Author:
Reviewer #2: Comments and Suggestions for Authors
Vargas-Alarcón et al. report that CASP1 gene polymorphisms and BAT1-NFKBIL-LTA-CASP1 gene-gene interactions are associated with restenosis after coronary stenting.
Major comments
Methods
Genetic analysis
1.- In the era of data/driven research, why and how did the Authors choose to pick a small number of polymorphisms, instead of performing whole genome/exome sequencing analyses of the patients?
Answer: Dear reviewer, we agree with you, however, to carry out a genome/exome sequencing study, we should have highly expensive state-of-the-art laboratory equipment, different reagents, specialized software, trained personnel in massive data analysis, as well as funding to this type of study. Unfortunately, until now we do not have the infrastructure to carry out genome/exome studies in our institution, therefore, we decided to carry out a candidate gene association study, taking into account previous polymorphisms associated with some cardiovascular pathologies, such as acute coronary syndrome.
Genetic interactions analysis
2.- More details on this analysis would be needed. How is a gene-gene interaction (epistasis?) being defined? Are these estimates based on 5 polymorphisms only? How is the background effect (i.e. when no gene-gene interactions exist) being estimated?
How is a gene-gene interaction (epistasis?) being defined?
Answer: Agree with comments of the reviewer. In response to these points.
The multifactor dimensionality reduction (MDR) program evaluates statistical epistasis, i.e. non-additive gene-gene interactions between genotypes of different loci and quantitative differences of allele-specific effects. In addition, MDR does not assume any genetic model and it is useful for the study of a complex disease whose mode of genetic inheritance is unknown.
In this context, we modified the following phrase: “The genetic interactions between BAT1, NFKBIL, LTA, and CASP1 polymorphisms in patients with and without restenosis were performed by the multifactor dimensionality reduction (MDR) program; the algorithm which allows for the evaluation of the quality of gene-gene interaction [20].” BY
“The genetic interactions between BAT1 rs2239527 C/G, NFKBIL1 rs2071592 T/A, LTA rs1800683 G/A, CASP1 rs501192 A/G, and CASP1 rs580253 A/G polymorphisms in patients with and without restenosis were performed by the multifactor dimensionality reduction (MDR) program. This program is useful for the study of a complex diseases whose mode of genetic inheritance is unknown. In addition, it evaluates statistical epistasis, i.e. non-additive gene-gene interactions between genotypes of different loci, as well as the quantitative differences of allele-specific effects, randomly validating the internal data through training, testing, and validation analysis. The interactions between multilocus genotypes are estimated using p values: when is less than 0.05, means that they are associated with the disease [20].” In material methods section.
Are these estimates based on 5 polymorphisms only?
Answer: In response to this point, the results show in tables and result section are based on the five SNPs studied in this work. As far as we know, this is the first report that showed an association of the gene-gene interaction between the BAT1, NFKBIL1, LTA, and CASP1 SNPs with higher risk of the development restenosis after coronary stenting in our population. Nonetheless, several studies have been showed that the analysis of the gene-gene interactions between polymorphisms of different genes play an important role in development of some complex diseases such as CAD, rheumatoid arthritis, and hyperlipidemia.
In this context, we changed the following phrase: “On the other hand, in this study, we did not observe an association of the rs2239527 C/G, rs2071592 T/A, and rs1800683 G/A polymorphisms with risk of developing restenosis after coronary stenting. Nonetheless, we evaluated whether the interaction of the BAT1 rs2239527 C/G, NFKBIL1 rs2071592 T/A, and LTA rs1800683 G/A polymorphisms with the rs580253 A/G and rs501192 A/G poly-morphisms of CASP1,…” BY
“In the last decade, several studies have shown that the analysis of gene-gene interactions between different genes plays an important role in the risk of developing complex diseases such as coronary heart disease, rheumatoid arthritis, and hyperlipidemia. [Deng et al. 2020, Ramirez-Bello et al. 2020, Musameh et al. 2015, Zhang et al. 2019]. In this study, we evaluated whether the gene-gene interaction between BAT1 rs2239527 C/G, NFKBIL1 rs2071592 T/A, LTA rs1800683 G/A, CASP1 rs580253 A/G and CASP1 rs501192 A/G polymorphisms…”. In discussion section.
In addition, we included the following references.
- Deng GX, Yin RX, Guan YZ, Liu CX, Zheng PF, Wei BL, et al. (2020). Association of the NCAN-TM6SF2-CILP2-PBX4-SUGP1-MAU2 SNPs and gene-gene and gene-environment interactions with serum lipid levels. Aging (Albany NY). 12:11893-11913.
- Ramírez-Bello J, Fragoso JM, Alemán-Ávila I, Jiménez-Morales S, Campos-Parra AD, Barbosa-Cobos RE, Moreno J. (2020). Association of BLK and BANK1 Polymorphisms and interactions with Rheumatoid Arthritis in a Latin-American Population. Front Genet. 20;11:58
- Musameh MD, Wang WY, Nelson CP, Lluís-Ganella C, Debiec R, Subirana I, et al. (2015). Analysis of gene-gene interactions among common variants in candidate cardiovascular genes in coronary artery disease. PLoS One. 10:e0117684.
- Zhang QH, Yin RX, Chen WX, Cao XL, Wu JZ. (2019). TRIB1 and TRPS1 variants, G × G and G × E interactions on serum lipid levels, the risk of coronary heart disease and ischemic stroke. Sci Rep. 9: 2376.
How is the background effect (i.e. when no gene-gene interactions exist) being estimated?
Answer: In order to clarify this point.
The MDR is used for collapsing high-dimensional genetic data into a single dimension. In this case, we used MDR to generate graphical models of LTA, BAT1, NFKBIL1, and Casp1 SNPs. MDR is a nonparametric and genetic model-free data mining method, in which, multilocus genotypes are pooled into high-risk and low risk groups, effectively reducing the dimensionality of the genotype predictors from N dimensions to one dimension. Thus, multilocus genotype attribute can be evaluated for its ability to classify and predict disease status. In addition, the interaction between multilocus genotypes is estimated using p values, which, when they are less than 0.05, means that they are associated with the disease. In the absence of p-values with statistical significance (p>0.05), it shows that there are no gene-gene interactions associated with the disease.
In this context, we added the following phrase: “(algorithm which evaluate the gene-gene interaction)” BY “…,which evaluate statistical epistasis based in models of genetic interaction between genotypes of different loci.” In discussion section.
Analysis of the haplotypes
3.- More details on this analysis would be needed. Did the Authors estimate haplotypes based on 5 polymorphisms only? What reference population was being used? Were there any quality scores attached to these estimates?
Answer: In response to reviewer comment. Barrett et al., define as haplotype to set of genetic variations along a chromosome that tend to be inherited together because they are in close proximity. The reason why they are inherited together is because there is usually no crossing over or recombination between these markers, or different polymorphisms, being so close. So, a haplotype can refer to a combination of alleles on a single gene, or alleles on multiple genes. In this sense, we estimate haplotypes based on 5 SNPs only, according to the position in the chromosome.
On the other hand, Haploview is a software that provides the statistical calculation of linkage disequilibrium (LD, D'), as well as the haplotype patterns from primary genotypes, using the platform of Human Haplotype Map project (International HapMap Consortium, 2003). In addition, the quality scores (Hedricks multiallelic D’, LOD, and r2) were calculates using standard EM (Expectation-Maximization) algorithm.
In this context, we change the following phrase: “The linkage disequilibrium analysis (LD, D’) and haplotypes construction were performed using Haploview version 4.1 (Broad Institute of Massachusetts Institute of Technology and Harvard University, Cambridge, MA, USA).” By “Haplotype design, and linkage disequilibrium analysis (LD, D') were performed using Haploview version 4.1 (Broad Institute of Massachusetts Institute of Technology and Harvard University, Cambridge, MA, USA). This software analyzes the combination of alleles in a single gene, or alleles in multiple genes along a chromosome, that tend to be inherited together due to the proximity between them, providing the statistical calculation of the linkage disequilibrium (LD, D'), LOD and r squared, as well as possible haplotype patterns from primary genotypes, using the platform of the Human Haplotype Map project data [Barret et al. 2005].” In Material and method section.
Added the following reference:
- Barrett JC, Fry B, Maller J, Daly MJ. Haploview: analysis and visualization of LD and haplotype maps. Bioinformatics. 2005 Jan 15;21(2):263-265.
Results
4.- Table 2 should be re-made by re-arranging cells in a different way, e.g. with vs. restenosis should be the columns, each of which could be further subdivided into the genetic models tested; genetic variants should be presented as rows, without empty cells in between, i.e. Overall brought closer to the standard representation of SNPs and their associations in the fields, e.g. Table 2 here https://www.frontiersin.org/articles/10.3389/fphar.2021.586973/full
Answer: Agree with the reviewer comment, we re-arranging cells, and changed the Table 2 now Table 3. In this Table, we changed title of table, and presentation of our data, according to inheritance models.
In this context, we changed the Table 2.
Table 2. Distribution of CASP1 polymorphisms in patients with and without restenosis.
|
|
|
Genotype Frequency |
|
MAF |
Model |
OR (95%CI) |
pC |
|
CASP1 G+7/in6A |
rs501192 |
|
|
|
|
|
|
|
With Restenosis |
GG |
GA |
AA |
A |
|
|
|
|
(n=66) |
25 (0.379) |
26 (0.394) |
15 (0.227) |
0.424 |
Co-dominant |
3.11 (1.15-8.40) |
0.078 |
|
|
|
|
|
|
Dominant |
1.59 (1.00-3.13) |
0.169 |
|
Without Restenosis |
76 (0.497) |
59 (0.386) |
18 (0.118) |
0.310 |
Recessive |
2.73 (1.10-6.81) |
0.031 |
|
(n=153) |
|
|
|
|
Over-dominant |
0.93 (0.48-1.81) |
0.839 |
|
|
|
|
|
|
Log-additive |
1.65 (1.02-2.65) |
0.039 |
|
|
|
|
|
|
|
|
|
|
CASP1 Exon 6 A10370-G |
rs580253 |
|
|
|
|
|
|
|
With Restenosis |
GG |
GA |
AA |
A |
|
|
|
|
(n=66) |
22 (0.333) |
26 (0.394) |
18 (0.273) |
0.469 |
Co-dominant |
5.38 (1.99-14.6) |
0.003 |
|
|
|
|
|
|
Dominant |
2.12 (1.06-4.23) |
0.031 |
|
Without Restenosis |
|
|
|
|
Recessive |
4.32 (1.76-10.6) |
0.001 |
|
(n=153) |
77 (0.503) |
58 (0.379) |
18 (0.118) |
0.307 |
Over-dominant |
0.91 (0.46-1.76) |
0.769 |
|
|
|
|
|
|
Log-additive |
2.16 (1.33-3.52) |
0.001 |
OR, odds ratio; CI, confidence interval; MAF, minor allele frequency; pC, p-value. The p-values were calculated by the logistic regression analysis. ORs were adjusted for age, gender, blood pressure, BMI, glucose, total cholesterol, HDL-C, LDL-C, triglycerides, hypertension, T2DM, dyslipidemia, and smoking. We also adjusted these ratios for the angiographic characteristics, such as unstable angina, stable angina, drug eluting stent (DES), bare metal stent (BMS), diameter smaller, and stent length.
By
Table 3. The inheritance models analysis of CASP1 polymorphisms in patients with and without restenosis.
|
SNP |
Model |
Genotype |
With Restenosis n=66 (n(%)) |
Without Restenosis n=219 (n(%)) |
OR (95%CI) |
pC |
|
rs501192 |
Co-dominant |
GG GA AA |
25 (0.379) 26 (0.386) 15 (0.227) |
76 (0.497) 59 (0.386) 18 (0.118) |
3.11 (1.15-8.40) |
0.078 |
|
|
Dominant |
GG GA + AA |
25 (0.379) 41 (0.621) |
76 (0.497) 77 (0.503) |
1.59 (0.81-3.13) |
0.169 |
|
|
Recessive |
GG + GA AA |
51 (0.773) 15 (0.227) |
135 (0.882) 18 (0.118) |
2.73 (1.10-6.81) |
0.031 |
|
|
Over-dominant |
GG + AA GA |
40 (0.606) 26 (0.394) |
94 (0.614) 59 (0.386) |
0.93 (0.48-1.81) |
0.839 |
|
|
Log-additive |
-------- |
--------- |
----------- |
1.65 (1.02-2.65) |
0.039 |
|
rs580253 |
Co-dominant |
GG GA AA |
22 (0.333) 26 (0.394) 18 (0.273) |
77 (0.503) 58 (0.379) 18 (0.118) |
5.38 (1.99-14.6) |
0.003 |
|
|
Dominant |
GG GA + AA |
22 (0.333) 44 (0.667) |
77 (0.503) 76 (0.497) |
2.12 (1.06-4.23) |
0.031 |
|
|
Recessive |
GG + GA AA |
48 (0.727) 18 (0.273) |
135 (0.882) 18 (0.118) |
4.32 (1.76-10.6) |
0.001 |
|
|
Over-dominant |
GG + AA GA |
40 (0.606) 26 (0.394) |
95 (0.621) 58 (0.379) |
0.91 (0.46-1.76) |
0.769 |
|
|
Log-additive |
-------- |
--------- |
----------- |
2.16 (1.32-3.52) |
0.001 |
Abbreviations: OR, odds ratio; CI, confidence interval; MAF, minor allele frequency; pC, p-value. The p-values were calculated by the logistic regression analysis. ORs were adjusted for age, gender, blood pressure, BMI, glucose, total cholesterol, HDL-C, LDL-C, triglycerides, hypertension, T2DM, dyslipidemia, and smoking. We also adjusted these ratios for the angiographic characteristics, such as unstable angina, stable angina, drug eluting stent (DES), bare metal stent (BMS), diameter smaller, and stent length.
5.- Table3 is currently also very hard to understand, each haplotype should represent one row for clarity, and empty cells should be avoided
Answer: Agree with the reviewer comment, we re-arranging cells, and changed the Table 3 now Table 4.
In this context, we changed the Table 3.
Table 3. Distribution of the haplotypes between the BAT1 -23C/G, NFKBIL1 -63T/A, and LTA -162G/A polymorphisms and the CASP1 G+7/in6A and CASP1 A10370-G polymorphisms in the study groups.
|
Haplotype |
|
|
With restenosis (n=66) |
Without restenosis (n=153) |
pC |
|
BAT1 -23C/G |
NFKBIL1_-63T/A |
LTA -162G/A |
Hf |
Hf |
|
|
C |
T |
G |
0.628 |
0.608 |
0.759 |
|
G |
A |
A |
0.311 |
0.366 |
0.316 |
|
C |
T |
A |
0.023 |
0.010 |
0.535 |
|
|
|
|
|
|
|
|
Haplotype |
|
|
Hf |
Hf |
|
|
CASP1 G+7/in6A |
CASP1 A10370-G |
|
|
|
|
|
G |
G |
|
0.461 |
0.689 |
1X10-4 |
|
A |
A |
|
0.355 |
0.306 |
0.371 |
|
G |
A |
|
0.114 |
0.001 |
<0.001 |
|
A |
G |
|
0.069 |
0.004 |
1X10-3 |
Abbreviations: Hf, Haplotype frequency; pC = p-corrected. The polymorphism order in the haplotypes is according to the positions in the chromosome.
BY
Table 4. Distribution of the haplotypes between the BAT1 rs2239527 C/G, NFKBIL1 rs2071592 T/A, and LTA 1800683 A/T polymorphisms, and the CASP1 rs501192 A/G and CASP1 rs580253 A/G polymorphisms in the study groups.
|
SNP |
With restenosis n=66 |
Without restenosis n=153 |
|
|
rs2239527 - rs2071592 -rs1800683 |
Hf (%) |
Hf (%) |
p |
|
CTG |
0.628 |
0.608 |
0.759 |
|
GAA |
0.311 |
0.366 |
0.316 |
|
CTA |
0.023 |
0.010 |
0.535 |
|
rs501192 - rs580253 |
Hf (%) |
Hf (%) |
p |
|
GG |
0.461 |
0.689 |
1X10-4 |
|
AA |
0.355 |
0.306 |
0.371 |
|
GA |
0.114 |
0.001 |
<0.001 |
|
AG |
0.069 |
0.004 |
1X10-3 |
Abbreviations: Hf, Haplotype frequency; pC = p-corrected. The polymorphism order in the different haplotypes is according to the positions in the chromosome. (rs2239527 - rs2071592 - rs1800683 chromosome 6q21.3, and rs501192-rs580523-chromosome 11q23).
6.- Interaction analysis and Table 4. This part is not clear at all. What does the number of factors represent? What do training and testing accuracy stand for? How was it calculated? How and with which data was the training performed?
What does the number of factors represent?
Answer: We apologize for this error; the correct word is “Number of interactions associated” and not number of factors. In this sense, we improve the Table 4 now Table 5 in the new manuscript.
What do training and testing accuracy stand for? How was it calculated? How and with which data was the training performed?
Answer: In order to clarify these points. We added the following phrase which help to understand better the specific part related with training, testing accuracy and cross-validation
“The MDR is mathematical model based on algorithms that can make predictions. This program implements the training set for all possible combinations of loci and the x models with the highest balanced accuracy are retained for evaluation in the testing set. After, MDR performs on all x models in the testing set and the best model for each level of interaction is preserved for evaluation of predictive ability in the validation set. Finally, one model is chosen to maximize balanced accuracy in the validation set [Winham et al. 2010].
Added the following reference:
- Winham SJ, Slater AJ, Motsinger-Reif AA. (2010). A comparison of internal validation techniques for multifactor dimensionality reduction. BMC Bioinformatics. 11:394.
On the other hand, we added the following sentence: “However, it is important to mention that the polymorphisms rs2239527 C/G, rs2071592 T/A and rs1800683 G/A individually showed no evidence of association with restenosis after coronary stenting. Nonetheless, when we analyzed the gene-gene interactions between the five SNPs of the four genes, interestingly, several of them showed a strong association with this pathology, increasing the OR values dramatically.” In discussion section.
Discussion
7.- „In the present study, we analyzed four genes and five polymorphisms“ Why only four genes and five polymorphisms, and exactly these genes and polymorphisms?
Answer: In order to clarified this point. The objective primordial of this study was to establish whether the rs2239527 C/G, rs2071592 T/A, rs1800683 G/A, rs501192 A/G, and rs580253 A/G polymorphisms are associated with the susceptibility to development restenosis after coronary stent placement in a cohort patients with CAD. Nonetheless, this objective was based on previous reports that showed association of these SNPs with different cardiovascular diseases, including two reports made by our study group.
In this context, we changed the following phrase: “In the present study, we analyzed four genes and five polymorphisms (BAT1 rs2239527 C/G, NFKBIL1 rs2071592 T/A, LTA rs1800683 G/A, CASP1 rs501192 A/G, and CASP1 rs580253 A/G) in order to establish their role in the genetic susceptibility to restenosis after coronary stenting. We found…” By “Recent studies have shown that the polymorphisms (BAT1 rs2239527 C/G, NFKBIL1 rs2071592 T/A, LTA rs1800683 G/A, CASP1 rs501192 A/G, and CASP1 rs580253 A/G) are associated with the presence ACS, myocardial infarction, and CAD [5-8,15].”
8.- Why did you perform gene-gene interactions, i.e. Why did you assume that the 5 SNPs in 4 genes should be „interacting“ ? What is the biological rationale beyond that?
9.- What is the medical utility of the 5 SNPs in 4 genes and the „interactions“ between them?
Answer: In response to the points 8 and 9.
The MDR program, is a software that identifies statistical gene-gene interactions, this means that it does not identify epistasis between genes in a biological sense. In this sense, to determine if there is a gene-gene interaction from the biological point of view, functional studies in cell lines are required to confirm the statistical findings. Nonetheless, the interactions between the genotypes of these 5 SNPs in these 4 genes, associated with risk of developing restenosis after coronary stenting, are important from the genetic and statistical point of view. However, we are aware that these data still lack medical utility. Because we cannot tell the patient that with these interactions between these genes they will present a more severe disease or will relapse faster than other patients, etc.
In this context, we added the following phrase: “Nonetheless, we considered that these findings should be taken with caution. Because the gene-gene interactions only are associations statistical between different genes identified by the MDR program, and not known whether these gene-gene interactions could play a biological relevant role.” In discussion section.
10.- „The association of these polymorphisms with cardiovascular and other diseases in different populations is scarce and controversial“ Indeed, as the Authors state „additional studies in a larger number of individuals have to be undertaken.“ Would it be possible to get access to an independent validation cohort?
Answer: Thanks for the comment. But we not have a larger number of individuals (patients or/and controls), and not have access some other group of individuals. Nonetheless, we previously reported the same polymorphisms (BAT1 rs2239527 C/G, NFKBIL1 rs2071592 T/A, LTA rs1800683 G/A, CASP1 rs501192 A/G, and CASP1 rs580253 A/G) in patients with acute coronary syndrome ACS and healthy controls. In these studies, we reported the association of the BAT1 rs2239527 C/G, CASP1 rs501192 A/G, and CASP1 rs580253 A/G polymorphisms with presence of acute coronary syndrome [Immunol Res. 2017; 65: 862-868, and Rev Invest Clin. 2020; 72: 19-24]. These works served as the basis for the objective of this study and were cited in the same.
In this context. We think that the data of the healthy individuals could be serve as a reference of an independently group without coronary artery disease. We included in a supplementary Table, the allele and genotype frequencies of these SNPs between CAD patients and healthy controls.
Table 2. Allele and genotype frequencies of BAT1 rs2239527 C/G, NFKBIL1 rs2071592 T/A, LTA rs1800683 G/A, CASP1 rs501192 A/G, and CASP1 rs580253 A/G genes polymorphisms in CAD patients and healthy controls
|
Polymorphic site (rsID-number) |
CAD n=219 (n(%)) |
Controls n=617 (n(%)) |
*p |
|
rs2239527 C/G |
|
|
|
|
Allele |
|
|
|
|
C |
278 (63.4) |
810 (65.6) |
NS |
|
G |
160 (36.5) |
424 (34.3) |
|
|
Genotype |
|
|
|
|
CC |
87 (39.7) |
272 (44.1) |
|
|
CG |
104 (47.4) |
266 (43.1) |
NS |
|
GG |
28 (12.8) |
79 (12.8) |
|
|
rs2071592 T/A |
|
|
|
|
Allele |
|
|
|
|
T |
281 (64.1) |
811 (65.7) |
NS |
|
A |
157 (35.8) |
423 (34.2) |
|
|
Genotype |
|
|
|
|
TT |
89 (40.6) |
268 (43.4) |
|
|
TA |
103 (47.0) |
275 (44.6) |
NS |
|
AA |
27 (12.3) |
74 (12.0) |
|
|
rs1800683 G/A |
|
|
|
|
Allele |
|
|
|
|
G |
275 (62.7) |
794 (64.3) |
NS |
|
A |
163 (37.2) |
440 (35.6) |
|
|
Genotype |
|
|
|
|
GG |
86 (39.2) |
260 (42.1) |
|
|
GA |
103 (47.0) |
274 (44.4) |
NS |
|
AA |
30 (13.7) |
83 (13.5) |
|
|
rs501192 G/A |
|
|
|
|
Allele |
|
|
|
|
G |
287 (65.5) |
823 (67.5) |
NS |
|
A |
151 (34.4) |
395 (32.4) |
|
|
Genotype |
|
|
|
|
GG |
101 (46.1) |
277 (45.5) |
|
|
GA |
85 (38.8) |
269 (44.2) |
0.040 |
|
AA |
33 (15.0) |
63 (10.3) |
|
|
rs580253 A/G |
|
|
|
|
Allele |
|
|
|
|
G |
282 (64.4) |
821 (67.4) |
NS |
|
A |
156 (35.6) |
397 (32.5) |
|
|
Genotype |
|
|
|
|
GG |
99 (45.2) |
275 (45.2) |
|
|
GA |
84 (38.3) |
271 (44.5) |
0.011 |
|
AA |
36 (16.4) |
63 (10.3) |
|
Data are shown as n and frequency. *chi-square test.
In addition, we included the following phrase: “As a comparison group, we included 625 healthy individuals with neither symptoms nor previous diagnosis of cardiovascular or systemic disease previously reported (Immunol Res. 2017; 65: 862-868, and Rev Invest Clin. 2020; 72: 19-24)”. In material and methods section.
On the other hand, we added the following phrase: “Genotype frequencies in the polymorphic sites were in Hardy-Weinberg equilibrium. In a first analysis, the genetic distribution of the rs2239527 C/G, rs2071592 T/A, and rs1800683 G/A, polymorphisms were similar in patients with CAD and healthy individuals. Nonetheless, the genotype frequencies of the rs580253 A/G and rs501192 A/G SNPs showed significant differences between CAD patients and healthy individuals (p < 0.05) (supplementary Table)”. In result section.
In addition, we replaced the following phrase: “Under recessive and additive models,…” By “In addition, when we realized the analysis of the rs580253 A/G and rs501192 A/G polymorphisms comparing patients with and without restenosis, we showed that under recessive and additive models,…” In result section.
On the other hand, we replaced the following phrase: “In addition, the association of these polymorphisms with cardiovascular and other diseases in different populations is scarce and controversial. For example,…” BY “In line with our data,…”
In addition, we changed the following phrase: “We consider that further studies with a greater number of individuals and different ethnic origins are needed to explain the true role of CASP1 polymorphisms in the risk of developing restenosis.” BY “Considering our data and the distribution of the alleles and genotypes of the rs501192 A/G, and rs580253 A/G polymorphisms in our population, we propose that additional studies in other populations with different ethnic origin could help define the true role of these polymorphisms as markers of risk in the developing restenosis after coronary stenting.” In discussion section.
Minor comments
11.- Please, remove „.“ from the title and the listing of authors
Answer: This was remove in new version of manuscript.
Abstract
12.- „Five gene polymorphisms (BAT1 -23C/G, NFKBIL1_-63T/A, LTA -162G/A, CASP1 G+7/in6A, and A10370-G)“ please use the rsID numbers of the respective SNVs throughout the manuscript
Answer: Agree with the reviewer comment, we changed the nomenclature of the polymorphisms (BAT1 -23C/G, NFKBIL1_-63T/A, LTA -162G/A, CASP1 G+7/in6A, and A10370-G) by the rsID numbers along of new version of manuscript.
13.- „The interactions gene-gene between BAT1-NFKBIL1-LTA-CASP1“ correct to „gene-gene interactions“
Answer: Agree with the reviewer comment, we apologize for this typographical error. This error was corrected in the new version of the manuscript.
Round 2
Reviewer 1 Report
The authors presented satisfactory responses to the comments and made corresponding changes to the text. After the necessary editorial corrections the paper could be published.
Answer: Thanks.
Reviewer 2 Report
Dear Authors,
Many thanks! It seems that all of my comments that could be addressed have been addressed, i.e. I am aware of the fact that performing WGS/WES in a 10x larger cohort may not be practically doable.
Kind regards,
The Reviewer
Answer: Thanks.